# Development and Validation of a Difficulty Scoring System for Laparoscopic Liver Resection to Treat Hepatolithiasis

**DOI:** 10.3390/medicina58121847

**Published:** 2022-12-15

**Authors:** Yeongsoo Jo, Jai Young Cho, Ho-Seong Han, Yoo-Seok Yoon, Hae Won Lee, Jun Suh Lee, Boram Lee, Eunhye Lee, Yeshong Park, MeeYoung Kang, Junghyun Lee

**Affiliations:** Department of Surgery, Seoul National University Bundang Hospital, Seoul National University College of Medicine, Seongnam-si 13590, Republic of Korea

**Keywords:** difficulty scoring system, laparoscopic liver resection, hepatolithiasis

## Abstract

*Background and Objectives*: A difficulty scoring system was previously developed to assess the difficulty of laparoscopic liver resection (LLR) for liver tumors; however, we need another system for hepatolithiasis. Therefore, we developed a novel difficulty scoring system (nDSS) and validated its use for predicting postoperative outcomes. *Materials and Methods*: This was a retrospective study. We used clinical data of 123 patients who underwent LLR for hepatolithiasis between 2003 and 2021. We analyzed the data to determine which indices were associated with operation time or estimated blood loss (EBL) to measure the surgical difficulty. We validated the nDSS in terms of its ability to predict postoperative outcomes, namely red blood cell (RBC) transfusion, postoperative hospital stay (POHS), and major complications defined as grade ≥IIIa according to the Clavien–Dindo classification (CDC). *Results*: The nDSS included five significant indices (range: 5–17; median: 8). The RBC transfusion rate (*p* < 0.001), POHS (*p* = 0.002), and major complication rate (*p* = 0.002) increased with increasing nDSS score. We compared the two groups of patients divided by the median nDSS (low: 5–7; high: 8–17). The operation time (210.7 vs. 240.7 min; *p* < 0.001), EBL (281.9 vs. 702.6 mL; *p* < 0.001), RBC transfusion rate (5.3% vs. 37.9%; *p* < 0.001), POHS (8.0 vs. 13.3 days; *p* = 0.001), and major complication rate (8.8% vs. 25.8%; *p* = 0.014) were greater in the high group. *Conclusions*: The nDSS can predict the surgical difficulty and outcomes of LLR for hepatolithiasis and may help select candidates for the procedure and surgical approach.

## 1. Introduction

The difficulty of surgical techniques is somewhat subjective and can be influenced by patient characteristics, disease severity, surgical equipment, type of surgery, and the surgeon’s experience [1]. Many scoring systems for surgical procedures have been proposed, including difficulty scores for laparoscopic cholecystectomy [2] and spinal anesthesia [3], the complexity of endotracheal intubation [4], and for predicting the complications of ophthalmological surgery [5]. Such systems can reveal a road map for young surgeons who are learning surgical techniques, via a step-by-step training regime [6] and can help surgeons provide patients with better information about the predicted risk of the procedure [7]. Scoring systems can also be used to make unbiased comparisons of cases of various difficulties among surgeons [8].

Laparoscopic liver resection (LLR) has shown impressive developments in the field of liver surgery in the last few decades [9,10]. In an effort to measure operative difficulty and generate a roadmap for surgeons advancing from simple to highly technical LLR, a difficulty scoring system (DSS) was developed to assess the difficulty of LLR for liver tumors [11]. The resulting DSS was determined based on the extent of liver resection, tumor location, tumor size, proximity to major vessels, hand-assisted laparoscopic surgery (HALS) or hybrid surgery, and liver function [12,13].

For hepatolithiasis, also known as intrahepatic duct (IHD) stones, hepatectomy is a safe and definitive treatment to treat diseased IHD. However, unlike liver tumors, non-anatomical resection is not recommended because of the IHD’s distribution, which can be easily distorted by stones or combined atrophy of liver parenchyma. Some indices used in the published DSS cannot be applied to LLR for hepatolithiasis, particularly proximity to major vessels and tumor size. Furthermore, HALS or hybrid surgery has not been performed for hepatolithiasis in recent years. Therefore, we developed a novel DSS for LLR to treat hepatolithiasis [14].

## 2. Materials and Methods

### 2.1. Study Design

This was a single-center retrospective study. We reviewed the clinical data for 138 patients who underwent LLR for hepatolithiasis between June 2003 and April 2021 at Seoul National University Bundang Hospital, Seongnam, South Korea. We excluded 15 patients who either underwent combined surgery at the same time (*n* = 8) or were diagnosed with malignant tumors postoperatively (*n* = 7); therefore, their surgical records and data were not appropriate to calculate the novel difficulty scoring system (nDSS) for LLR. We had nine patients who were converted to open approach, but six of them were confirmed with malignant disease and the others underwent combined surgery. These factors already corresponded to the exclusion criteria of this study, so we did not analyze open conversion cases. Accordingly, we analyzed data for 123 patients. To estimate the surgical difficulty, we calculated scores using indices that were significantly associated with operation time and estimated blood loss (EBL), which are generally thought to be the key markers for the difficulty of LLR. We validated the nDSS in terms of its ability to predict postoperative outcomes, namely red blood cell (RBC) transfusion, postoperative hospital stay (POHS), and major complications defined as grade ≥IIIa according to the Clavien–Dindo classification (CDC). We also divided the patients into two groups according to their nDSS (low, 5–7; high, ≥8) and evaluated the short-term outcomes to simplify the surgical decisions. This study was approved by the hospital’s Institutional Review Board (B-2208-773-105).

### 2.2. Surgical Techniques

The LLR techniques used at our institution are described in a previous report [15]. If remnant duct stones were suspected, intraoperative bile duct exploration was performed [16]. After dividing the liver parenchyma, the duct of the section or hemiliver was isolated. If the surgeon presumed that stones were close to the resection plane, the duct was divided with endo scissors. The stones near the open duct were extracted and the duct was closed using intracorporeal sutures. To detect any remnant stones, further exploration was performed via intraoperative choledochoscopy through the open duct before closing the duct [17]. Similarly, if common bile duct stones were suspected, the surgeon performed intraoperative common bile duct exploration in the same way. In this study, the operations were performed by five different surgeons and all of them had sufficient experiences of LLR, at least 30 cases each.

### 2.3. Definitions

Hepatolithiasis could be associated with IHD stricture, which can be observed by magnetic resonance cholangiopancreatography (MRCP; Figure 1A,B). The proximity to the bifurcation was defined as the distance between the distal end of the stricture and the confluence of the IHD affected by the stricture. We used this definition because, when planning anatomical liver resection, it is very important to draw the resection plane distal to the IHD confluence to avoid bile duct injury. If the distance is <1 cm, it might be more difficult to perform LLR properly without causing bile duct injury. Figure 1 shows two examples of MRCP depicting the proximity to the bifurcation. As an example, in a patient who was undergoing left hemihepatectomy, we drew the imaginary resection line just on the confluence of the left hepatic duct and common hepatic duct, and measured the distance between the imaginary resection line and the distal end of the left hepatic duct stricture (Figure 1A). Similarly, in a patient who was undergoing right hemihepatectomy, we measured the distance between the imaginary resection line and the distal end of the right hepatic duct stricture (Figure 1B). We reviewed all MRCP images from each patient and measured the distance in this way.

### 2.4. Statistics

All data were analyzed using SPSS version 20.0 for Windows (SPSS, Chicago, IL, USA). Continuous variables were compared using Student’s *t* test. Categorical variables were compared using the χ2 test or Fisher’s exact test. We also performed univariate and multivariable logistic regression analyses. In multivariate logistic regression analysis, we selected all significant variables from univariate logistic regression analysis. In all tests, a *p* value of ≤0.05 was regarded as significant.

## 3. Results

### 3.1. Patient Characteristics

We analyzed data for 46 males (37.4%) and 77 females (62.6%) (Table 1). Their mean age was 60 years and the mean BMI was 23.7 kg/m^2^. Twenty-nine patients (23.6%) had a history of upper abdominal surgery, including hepatobiliary and pancreatic surgery or gastroduodenal surgery, and some of them underwent multiple procedures. The prior procedures were cholecystectomy in most of these patients (*n* = 22), extrahepatic bile duct surgery (*n* = 9), pancreaticoduodenectomy (*n* = 3), hepatectomy (*n* = 2), and gastrectomy (*n* = 1). The LLR was classified into two types: left lateral sectionectomy (*n* = 43; 35.0%) and major hepatectomy (*n* = 80; 65.0%). The resection side was also classified into two groups: left (*n* = 106; 86.2%) and right (*n* = 17; 13.8%). Atrophy of the liver parenchyma was observed in 64 patients (52.0%). Fifty-one patients (41.5%) underwent IHD exploration, as described in the Methods (Surgical techniques). The hepatolithiasis was in close proximity to the bifurcation in 63 patients (51.2%). Among the patients who were corresponded to the inclusion criteria, nobody underwent biliary reconstructions with Roux-en-Y hepaticojejunostomy.

### 3.2. Surgical Outcomes

The median operation time was 260 min and the median EBL was 300 mL. In total, 28 patients (22.8%) received RBC transfusion and the median POHS was 8 days. Eleven patients (8.9%) had remnant stones. Twenty-two patients (17.9%) experienced severe postoperative complications with CDC grade of ≥IIIa (Table 2). To determine which factors were associated with the surgical difficulty, we performed a logistic regression analysis using operation time longer than the median (260 min) as the dependent variable. In the multivariable analysis, four variables were significantly associated with this outcome: resection type (odds ratio [OR]: 3.984; 95% confidence interval [CI]: 1.596–9.947; *p* = 0.003), resection side (OR: 4.173; 95% CI: 1.018–17.104; *p* = 0.047), intraoperative bile duct exploration (OR: 3.891; 95% CI: 1.678–9.021; *p* = 0.002), and proximity to the bifurcation (OR: 2.683; 95% CI: 1.1487–6.269; *p* = 0.023) (Table 3). We also performed logistic regression using EBL greater than the median (300 mL) as the dependent variable. In the multivariable analysis, three variables were significantly associated with this outcome: resection side (OR: 16.209; 95% CI: 2.007–130.901; *p* = 0.009), intraoperative bile duct exploration (OR: 2.812; 95% CI: 1.225–6.455; *p* = 0.015), and history of upper abdominal surgery (OR: 3.976; 95% CI: 1.408–11.231; *p* = 0.009) (Table 3).

### 3.3. Development of the nDSS and Associations between nDSS and Short-Term Outcomes

Five variables, including those that overlapped both multivariable regression models, were included in the nDSS: resection type, resection side, intraoperative bile duct exploration, proximity to the bifurcation, and history of upper abdominal surgery. When a patient had some factors, we assigned points to each factor according to their odds ratios. We multiplied the odds ratios from the results of multivariable analyses based on operation time and EBL, and extracted the square root of them, and rounded off to the nearest whole number; for example, when it comes to resection side and if it is the right side, the point is 8 ($\approx \sqrt{4.173\times 16.209}$). If the factor had a significance on only one dependent variable, the odds ratio from the other side was considered as 1; for example, when it comes to proximity to the bifurcation and if IHD stricture is <1 cm from the bifurcation, the point is 2 ($\approx \sqrt{2.683\times 1})$. If a patient did not have some specific factors, we assigned 1 point to each factor. In conclusion, each factor had their own points (resection type, 2; resection side, 8; intraoperative bile duct exploration, 3; proximity to the bifurcation, 2; history of upper abdominal surgery, 2), which are summed to provide the nDSS with a possible range of 5 to 17 points. However, in contrast to common perception, liver parenchyma atrophy was not a significant variable in either model, and this was excluded from the nDSS.

Scatter plots for operation time and EBL versus nDSS are shown in Figure 2. Both graphs showed that operation time and EBL tended to increase with greater nDSS. To evaluate the use of nDSS for predicting short-term outcomes, we performed univariate logistic regression analyses for five variables each, and three variables showed significance with increasing nDSS: RBC transfusion rate (OR: 1.383; 95% CI: 1.186–1.614; *p* < 0.001), POHS ≥ 8 days (OR: 1.307; 65% CI: 1.107–1.544; *p* = 0.002), and CDC grade ≥IIIa (OR: 1.267; 95% CI: 1.092–1.470; *p* = 0.002) (Table 4). Remnant stones (OR: 0.928; 95% CI: 0.722–1.194; *p* = 0.564) and recurrent stones (OR: 0.948; 95% CI: 0.686–1.310; *p* = 0.746) were not significantly associated with nDSS.

To understand whether the nDSS can be useful for treatment decisions, including surgical approach and patient selection, we divided the patients into two groups based on the median nDSS and compared the surgical outcomes between the two groups. The low group comprised patients with a score of 5–7 points and the high group comprised patients with a score of 8–17 points. As shown in Table 5, the low group had significantly better short-term outcomes than the high group for operation time ≥260 min (28.1% vs. 72.7%; OR: 6.833; *p* < 0.001), EBL ≥ 300 mL (42.1% vs. 77.3%; OR: 4.675; *p* < 0.001), RBC transfusion rate (5.3% vs. 37.9%; OR: 10.976; *p* < 0.001), POHS ≥ 8 days (35.1% vs. 66.7%; OR 3.700; *p* = 0.001), and CDC grade ≥IIIa (8.8% vs. 25.8%; OR: 3.608; *p* = 0.014), but no differences were found for remnant or recurrent stones.

## 4. Discussion

Although there are many different treatment modalities for hepatolithiasis, hepatectomy seems to be one of the most effective options because it can reduce the risk of recurrence and cholangiocarcinoma [18,19,20,21]. With advances in laparoscopic techniques and accumulating clinical evidence for better short-term outcomes and comparable long-term outcomes [22,23,24,25,26,27], LLR is increasingly being used for treating hepatolithiasis [28,29]. However, LLR for hepatolithiasis may be more technically challenging than for tumors because inflammation of the liver associated with hepatolithiasis leads to perihepatic adhesion and anatomical distortion [30]. Furthermore, parenchymal transection is often difficult because of parenchymal fibrosis and deformation of the IHD due to atrophic changes [16]. These factors could extend the operation time and increase the risk of postoperative complications. Moreover, intraoperative choledochoscopic evaluation of the remaining biliary tract is often required, and further prolongs the operation time and increases the surgical difficulty [16,31].

Several studies have developed a surgical DSS for LLR [11,12,13,32,33]. However, very few studies have evaluated the difficulty of LLR for hepatolithiasis. Here, we found that the surgical difficulty varies among patients undergoing the same LLR procedure and that the nDSS can be applied to LLR for hepatolithiasis. The surgical difficulty increases with nDSS (Figure 2) and the surgical outcomes are worse at higher nDSS (Table 4). Therefore, surgeons can use this system to predict the surgical difficulty and outcomes, and share the information with anesthesiologists, intensivists, hospitalists, nurses, and any other medical team members involved in the treatment, for appropriate pre-, intra-, and postoperative arrangements, such as preparation of blood transfusion, anesthetic drugs, surgical equipment, and intensive care. Furthermore, if the patients can be divided into low or high scores, based on the median score of 8 points, it is simpler to inform the patients about the likelihood of longer hospital stay or greater risk of postoperative complications, and to facilitate decisions on the surgical approach. Furthermore, because the nDSS is an unbiased tool that measures surgical difficulty quantitatively, it can be used to compare cases and determine which factor(s) may affect the surgery and postoperative outcomes. Thus, we believe the nDSS can be used as a roadmap for using LLR to treat hepatolithiasis.

We had a few unexpected results. One of them was about history of upper abdominal surgery. We defined upper abdominal surgery as hepatobiliary and pancreatic surgery or gastroduodenal surgery. If the patient had undergone very extensive surgery, for example, pancreaticoduodenectomy or hepatectomy, the surgeon might have decided to perform open surgery worrying about surgical difficulty. Otherwise, if the patient had undergone minor surgery, such as laparoscopic cholecystectomy, it might not have been a serious matter to go with laparoscopic surgery. That might have been one of the reasons that history of upper abdominal surgery was not a significant factor when it comes to operation time. However, we could not find any acceptable reasons to explain why it had a significant effect on EBL.

Another unexpected result was about bile duct exploration: even though it is not always difficult for experienced surgeons, once the procedure is performed, it is quite reasonable that surgery takes longer than that without the procedure. However, we could not find any explainable reasons of the result for why it had a significant effect on EBL as well. Otherwise, when it comes to resection type, it did not show any significant effect on EBL in contrast with common knowledge. Hence, further studies are warranted on these issues.

Regardless of the usefulness of nDSS, this study has some limitations to discuss. First, this was a single-center retrospective study with a risk of selection bias and other disadvantages inherent to such studies. For example, patients with severe liver parenchyma atrophy would not have been considered as candidates for laparoscopic surgery. If liver parenchyma atrophy is very severe, it could be very difficult to determine whether there is malignant tumor or not just based on preoperative image findings. Due to this reason, when liver parenchyma atrophy was very severe, we performed open surgery in case of achieving appropriate resection margins, performing lymph node dissection or hepaticojejunostomy according to intraoperative findings. This might explain why liver parenchyma atrophy was not a significant factor for surgical difficulty in our study, and that is the same with the matter of the ‘few unexpected results’ that we discussed before. Second, the nDSS was not associated with the remnant stone and recurrent stone rate. Of course, if the stones were located in both hemilivers, we resected the atrophied or more severe side of the liver and observed the patient prior to further resection. Accordingly, some IHD stones were intentionally left in situ (*n* = 4; 36.4%); this could affect the short-term outcome of the remnant stone rate. Considering the goal of surgery, it is important to perform further studies to determine the curability of this strategy. Third, because not all operations were performed by the same surgeons, even though the surgeons used near-identical techniques and had sufficient experiences of LLR (at least ≥30 cases each), differences in their surgical skill levels might affect the surgical difficulty and outcomes. Furthermore, we cannot ignore the evolution of laparoscopic equipment and devices during the study period. Finally, although some patients had chronic liver disease or liver cirrhosis, which could affect the difficulty or outcomes, we did not incorporate this factor due to limited data, and future studies should investigate the impact of these diseases.

## 5. Conclusions

In conclusion, we found that the surgical difficulty varies among patients undergoing LLR for hepatolithiasis. We know that more difficult surgical procedures carry greater risk of worse postoperative outcomes. The nDSS developed here can predict the surgical difficulty and short-term outcomes of LLR for hepatolithiasis. Furthermore, we expect the nDSS will also be useful for selecting candidate patients and deciding between laparoscopic or open surgery.

## Figures and Tables

**Figure 1 medicina-58-01847-f001:**
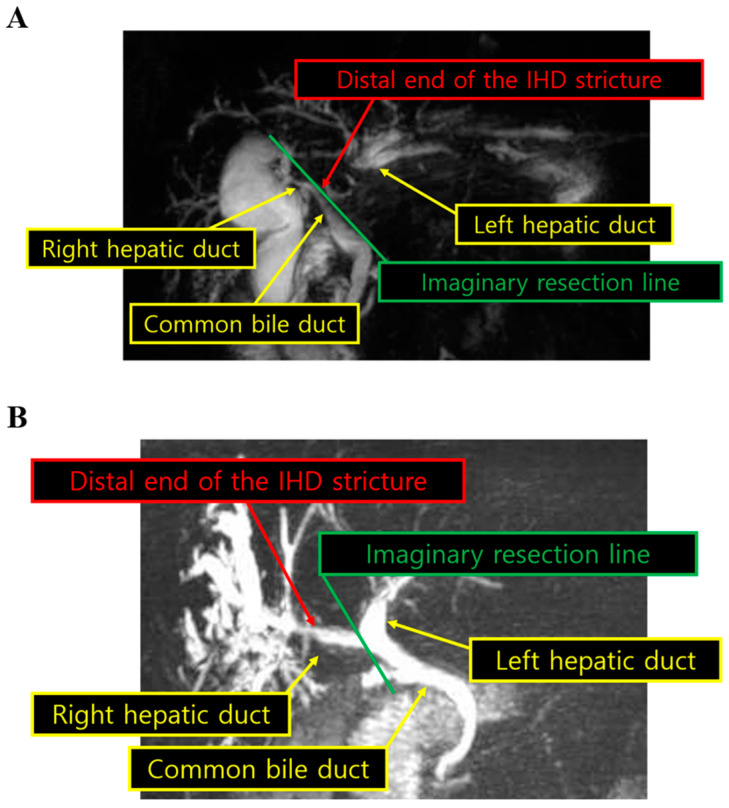
Preoperative magnetic resonance cholangiopancreatography. The proximity to the bifurcation was defined as the distance between the distal end of the stricture and the confluence of the IHD affected by the stricture. The straight line represents imaginary resection line, and we measured the distance between the imaginary resection line and the distal end of the stricture in representative patients undergoing left hemihepatectomy ((**A**); distance < 1 cm) and right hemihepatectomy ((**B**); distance ≥ 1 cm). IHD intrahepatic duct.

**Figure 2 medicina-58-01847-f002:**
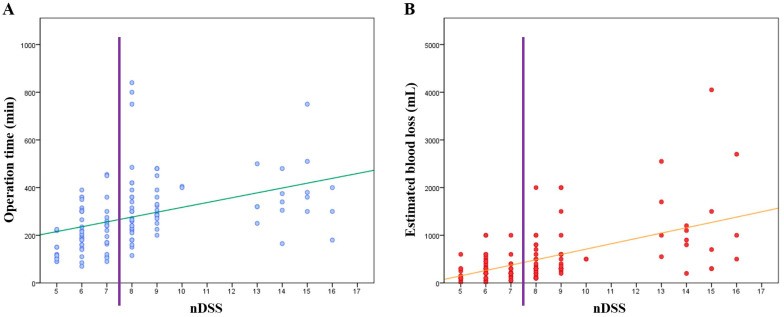
Scatter plots for nDSS versus operation time (**A**) and EBL (**B**). The vertical line splits the patients into two groups, low and high groups, and the oblique lines represent the correlations between the variables. nDSS modified difficulty scoring system.

**Table 1 medicina-58-01847-t001:** Preoperative characteristics.

Characteristics (*n* = 123)	Value
Age, years (mean)	59.98 ± 9.25
Sex, *n* (%)	
Male	46 (37.4%)
Female	77 (62.6%)
BMI, kg/m^2^ (mean)	23.67 ± 3.01
History of upper abdominal surgery, *n* (%)	29 (23.6%)
Resection type, *n* (%)	
Left lateral sectionectomy	43 (35.0%)
Major hepatectomy	80 (65.0%)
Resection side, *n* (%)	
Left hemiliver	106 (86.2%)
Right hemiliver	17 (13.8%)
Liver parenchyma atrophy, *n* (%)	64 (52.0%)
Bile duct exploration, *n* (%)	51 (41.5%)
IHD stricture <1 mm from the bifurcation, *n* (%)	63 (51.2%)

BMI body mass index, IHD intrahepatic duct.

**Table 2 medicina-58-01847-t002:** Surgical outcomes.

Variable	Value
Operation time (min)	Mean: 280.46 ± 141.63; median: 260
EBL (mL)	Mean: 507.64 ± 590.43; median: 300
RBC transfusion, *n* (%)	28 (22.8%)
RBC transfusion (mL)	Mean: 302.44 ± 784.01
POHS (days)	Mean: 10.85 ± 9.70, median: 8
Remnant stone, *n* (%)	11 (8.9%)
Recurrent stone, *n* (%)	6 (4.9%)
CDC grade ≥IIIa, *n* (%)	22 (17.9%)
Fluid collection, *n* (%)	10 (8.1%)
Biliary fistula, *n* (%)	6 (4.9%)
Pleural effusion, *n* (%)	2 (1.6%)
Biliary stricture, *n* (%)	1 (0.8%)
Septic shock, *n* (%)	1 (0.8%)
Pseudoaneurysm rupture, *n* (%)	1 (0.8%)
Wound complication, *n* (%)	1 (0.8%)

EBL estimated blood loss, RBC red blood cell, POHS postoperative hospital stay, CDC Clavien–Dindo classification.

**Table 3 medicina-58-01847-t003:** Logistic regression analysis for operation time ≥ 260 min and estimated blood loss ≥ 300 mL as the dependent variables.

	Operation Time ≥ 260 min						Estimated Blood Loss ≥ 300 mL					
	Univariate			Multivariable			Univariate			Multivariable		
Variables	OR	95% CI	*p* Value	OR	95% CI	*p* Value	OR	95% CI	*p* Value	OR	95% CI	*p* Value
Age ^1^	0.979	0.394–2.435	0.964	-	-	-	1.123	0.461–2.735	0.798	-	-	-
Sex ^2^	0.760	0.313–1.848	0.545	-	-	-	0.628	20.259–1.520	0.302	-	-	-
BMI ^3^	1.771	0.701–4.474	0.227	-	-	-	1.582	0.628–3.985	0.331	-	-	-
Resection type ^4^	4.479	1.702–11.785	**0.002**	3.984	1.596–9.947	**0.003**	1.759	0.740–4.183	0.201	-	-	-
Resection side ^5^	4.267	1.018–17.886	**0.047**	4.173	1.018–17.104	**0.047**	13.172	1.562–111.047	**0.018**	16.209	2.007–130.901	**0.009**
Liver parenchyma atrophy ^6^	0.744	0.312–1.770	0.504	-	-	-	0.622	0.267–1.452	0.272	-	-	-
Bile duct exploration ^7^	4.172	1.761–9.883	**0.001**	3.891	1.678–9.021	**0.002**	2.712	1.164–6.318	**0.021**	2.812	1.225–6.455	**0.015**
Proximity to the bifurcation ^8^	2.744	1.136–6.624	**0.025**	2.683	1.148–6.269	**0.023**	1.425	0.624–3.255	0.400	-	-	-
History of UAS ^9^	1.708	0.728–4.004	0.218	-	-	-	3.096	1.155–8.301	**0.025**	3.976	1.408–11.231	**0.009**

^1^ <65 vs. ≥65 years; ^2^ male vs. female; ^3^ <25 vs. ≥25 kg/m^2^; ^4^ LLS vs. major; ^5^ left vs. right; ^6^ no vs. yes; ^7^ no vs. yes; ^8^ ≥1 cm vs. <1 cm; ^9^ no vs. yes. *p* values in bold are statistically significant at ≤0.05. OR odds ratio, CI confidence interval, BMI body mass index, LLS left lateral sectionectomy, UAS upper abdominal surgery.

**Table 4 medicina-58-01847-t004:** Surgical outcomes based on the novel difficulty scoring system.

Variable	OR	95% CI	*p* Value
RBC transfusion	1.383	1.186–1.614	**<0.001**
POHS ≥ 8 days	1.307	1.107–1.544	**0.002**
CDC grade ≥ IIIa	1.267	1.092–1.470	**0.002**
Remnant stone	0.928	0.722–1.194	0.564
Recurrent stone	0.948	0.686–1.310	0.746

*p* values in bold are statistically significant at ≤0.05. OR odds ratio, CI confidence interval, EBL estimated blood loss, RBC red blood cell, POHS postoperative hospital stay, CDC Clavien–Dindo classification.

**Table 5 medicina-58-01847-t005:** Surgical outcomes in patients divided into high and low nDSS scores.

Variable	nDSS 5–7	nDSS ≥ 8	OR	*p* Value
Operation time, min (mean)	210.67	240.74	-	**<0.001**
Operation time ≥ 260 min, *n* (%)	16 (28.1%)	48 (72.7%)	6.833	**<0.001**
EBL, mL (mean)	281.93	702.58	-	**<0.001**
EBL ≥ 300 mL, *n* (%)	24 (42.1%)	51 (77.3%)	4.675	**<0.001**
RBC transfusion, *n* (%)	3 (5.3%)	25 (37.9%)	10.976	**<0.001**
POHS, days (mean)	8.00	13.32	-	**0.001**
POHS ≥ 8 days, *n* (%)	20 (35.1%)	44 (66.7%)	3.700	**0.001**
CDC grade ≥ IIIa, *n* (%)	5 (8.8%)	17 (25.8%)	3.608	**0.014**
Remnant stone, *n* (%)	4 (7.0%)	7 (10.6%)	1.572	0.543
Recurrent stone, *n* (%)	2 (3.5%)	4 (6.1%)	1.774	0.685

*p* values in bold are statistically significant at ≤0.05. nDSS modified difficulty scoring system, OR odds ratio, EBL estimated blood loss, RBC red blood cell, POHS postoperative hospital stay, CDC Clavien–Dindo classification.

## Data Availability

Data will be made available by the authors upon reasonable request.

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
