# Peer review of "Development and Validation of a Difficulty Scoring System for Laparoscopic Liver Resection to Treat Hepatolithiasis"

_medicina, 2022, doi:10.3390/medicina58121847_

Round 1

Reviewer 1 Report

I would like to congratulate the authors for presenting a very detailed and informative study results. The nDSS can predict the surgical difficulty and short-term outcomes of LLR for hepatolithiasis. Furthermore, it will also be useful for selecting candidate patients and deciding between laparoscopic or open surgery. I believe many surgeons will be interested in reading this paper.

Author Response

Thank you so much for your kind review of the article.

I really appreciate it.

Reviewer 2 Report

I carefully read the study by Yeongsoo Jo et al entitled: “Development and Validation of a Difficulty Scoring System for Laparoscopic Liver Resection to Treat Hepatolithiasis”.

The study is interesting and well structured. I have only minor comments

• It is not described how many patients were converted to open approach and why conversion  was not considered an outcome parameter

• It is not clear in how many cases a biliary reconstruction with Roux-en-Y hepaticojejunostomy had to be performed

• The first part of chapter 3.3 should be considered a “Method” and not a “Results”

• Major complications should be described in detail

Author Response

Point 1:  It is not described how many patients were converted to open approach and why conversion  was not considered an outcome parameter.

Response 1: We had nine patients who were converted to open approach, but six of them were confirmed with malignant disease and others underwent combined surgery. Those things were already corresponded to the exclusion criteria of this study, so we did not analyze open conversion cases.

Point 2: It is not clear in how many cases a biliary reconstruction with Roux-en-Y hepaticojejunostomy had to be performed.

Response 2: Among the patients who were corresponded to the inclusion criteria, nobody underwent biliary reconstructions with Roux-en-Y hepaticojejunostomy.

Point 3: The first part of chapter 3.3 should be considered a “Method” and not a “Results”

Response 3: We thought about exact the same way, but without the results of mutivariable analyses, it was not easy to explain it clearly.

Point 4: Major complications should be described in detail

Response 4: I'll add this information into the article (Table 2).